# Diversity and Experiences of Radiation Oncologists in Canada: A Survey of Gender Identity, Sexual Orientation, Disability, Race, Ethnicity, Religion, and Workplace Discrimination—A National Cross-Sectional Electronic Survey

**DOI:** 10.3390/curroncol32110643

**Published:** 2025-11-17

**Authors:** Amanda F. Khan, Stefan Allen, Ian J. Gerard, Rhys Beaudry, Glen Bandiera, David Bowes, Jolie Ringash, Reshma Jagsi, Jennifer Croke, Shaun K. Loewen

**Affiliations:** 1Department of Oncology, University of Calgary, Calgary, AB T2N 4N1, Canada; shaun.loewen@albertahealthservices.ca; 2Department of Radiation Oncology, University of Toronto, Toronto, ON M5T 1P5, Canada; stefanc.allen@nshealth.ca (S.A.); jolie.ringash@uhn.ca (J.R.); jennifer.croke@uhn.ca (J.C.); 3Division of Radiation Oncology, McGill University, Montreal, QC H4A 3T2, Canada; ian.gerard@mail.mcgill.ca; 4Mozell Family Statistical Analysis Core, University of Calgary, Calgary, AB T2N 1N4, Canada; rhys.beaudry@ucalgary.ca; 5Department of Medicine, University of Toronto, Toronto, ON M5S 1A8, Canada; glen.bandiera@utoronto.ca; 6Department of Radiation Oncology, Dalhousie University, Halifax, NS B3H 4R2, Canada; david.bowes@nshealth.ca; 7Department of Radiation Oncology, Emory School of Medicine, Atlanta, GA 30322, USA; rjagsi@emory.edu

**Keywords:** diversity, equity, workplace discrimination, Canadian, radiation oncologists, race, gender, harassment, ethnicity, age

## Abstract

A national cross-sectional electronic survey was designed to explore the demographics and experiences of discrimination among Canadian radiation oncologists (ROs), focusing on ethnicity, gender identity, disability, sexual orientation, and socioeconomic background. An ethics-approved bilingual survey was sent to all Canadian ROs (*n* = 598), with 42.5% of ROs completing the survey in full (254/598). Descriptive statistics, Chi-square tests, and content analysis were used. Most respondents were male, heterosexual, and aged 35–44. Racialized individuals made up 41.2%, though Black, Indigenous, and South Asian ROs were underrepresented. Only 20% of racialized ROs were women versus 49.3% among White ROs. While 75.4% reported job satisfaction, 42.1% experienced workplace discrimination, mainly from colleagues or patients. Racialized ROs reported twice the harassment rate of White ROs, and 45.2% felt unable to report it. The findings reveal significant discrimination, especially for racialized women, and highlight the urgent need for diversity initiatives, mentorship, and stronger institutional policies.

## 1. Introduction

A U.S. diversity assessment of radiation oncologists (ROs) found that women and individuals underrepresented in medicine (URiM) (including Black, Hispanic, and Indigenous people) remain significantly underrepresented in radiation oncology compared to other specialties [1]. URiM physicians report lower career satisfaction, fewer mentorship and promotion opportunities, and higher attrition rates than non-URiM peers [2]. Non-White physicians, particularly Black physicians and non-White women, experience higher rates of discrimination, negatively affecting career advancement, work environment, and health [3,4,5]. Those most affected by harassment and discrimination often lacked knowledge of reporting mechanisms, and fewer than 25% were satisfied with how reports were handled [4]. In Europe, the European Society for Radiotherapy and Oncology (ESTRO) found that minority radiation oncology professionals reported lower inclusion scores than non-minorities, with the lowest ratings among those identifying gender as their minority criterion [6].

Although no comparable diversity assessment exists for Canadian radiation oncologists, related fields show similar trends. The 2021 Canadian Organization of Medical Physicists (COMP) Equity, Diversity, and Inclusion Climate Survey found that only 36% of respondents were women and less than 1% were Indigenous, despite Indigenous people comprising 5% of Canada’s workforce [7]. Fewer women (58%) than men (70%) reported access to professional opportunities, and 17% (mostly women) reported experiencing sexual harassment in the past five years [7].

The American College of Radiology (ACR) Commission for Women and Diversity reported that radiation oncology training programs are less diverse than medical schools or other specialties [8]. They emphasized that a diverse workforce enhances patient care, communication, and outcomes by fostering cultural competence and trust [9,10,11].

Ethnicity also influences cancer outcomes [1,12]. U.S. Black men are twice as likely to die from prostate cancer as White men and are less likely to receive screening, even after adjusting for socioeconomic factors [13]. South Asian women in Ontario and African American and Hispanic women in the U.S. are more often diagnosed with advanced breast cancer [14,15]. Physician race and ethnicity also affect patient satisfaction: patients report greater comfort and trust when their physician shares their racial or ethnic background. The National Academy of Medicine’s *Unequal Treatment* report highlights how provider implicit bias contributes to racial disparities in healthcare [16]. Relatedly, in a U.S. study, Black, Hispanic, and Asian patients reported greater satisfaction with care if their doctor was of the same race or ethnic background than if their doctor was of a different background because of increased comfort and trust [11]. Recruiting and retaining a diverse physician workforce strengthens healthcare by improving cultural understanding, patient advocacy, and access to care [17,18].

The American College of Radiology (ACR) made numerous recommendations in its Diversity Report, including (i) advocacy and awareness through publications, (ii) the creation and dissemination of metrics, and (iii) the collection of data that quantifies existing disparities so that specific areas can be targeted through EDI initiatives [19]. To address these recommendations in Canada, this study was created with the following objectives: (a) to be the first study to analyze and evaluate the demographics of Canadian ROs, (b) to gauge Canadian RO career satisfaction, workplace culture, and lived experiences of discrimination and harassment, and (c) to recognize areas of improvement within the field.

## 2. Materials and Methods

### 2.1. Study Population

The study was designed as a cross-sectional electronic survey. Eligibility criteria included being a Canadian radiation oncologist currently practicing in a Canadian province. Participants were recruited via the Canadian Association of Radiation Oncology email registry and through an email sent by administrative leaders at all 52 cancer facilities in Canada with radiotherapy services. Trainees (including fellows) were excluded. Survey responses were collected online via the Qualtrics survey platform between July 2023 and December 2023 after three email reminders to participate were sent out over a period of 6 weeks.

### 2.2. Survey Tool Design

The 45-question survey was developed to gather anonymous cross-sectional data, as there was no validated survey specifically designed for radiation oncologists in regard to EDI. The survey creation process was informed by a literature review and input from equity, diversity, and sociodemographic survey experts across Canada and the U.S. to ensure questions were inclusive, culturally sensitive, and able-focused rather than disability-focused, and when appropriate, were drawn from validated survey sets in other fields of medicine. The concept of underrepresented and minoritized individuals was based on the AAMC’s definition of representation in medicine relative to the general population [20]. A pilot survey was conducted at our local institution to identify confusing wording, leading questions, and define clear inclusion and exclusion criteria to avoid non-representative responses.

Survey questions were screened for clarity, bias, and definitions for terms like sexual orientation, gender identity, race/ethnicity, and workplace concepts such as “culture of respect” and “discrimination” were provided. Explanations for specific questions were included to give participants context. Ethnic and cultural origin questions followed the 2021 Canadian Census format, allowing respondents to identify multiple origins. Racialized status was defined per the 2016 Canadian Census for comparability. Gender identity questions included definitions and allowed for multiple responses, including options like agender, non-binary, and transgender, with a self-identification option.

The survey had four sections: (i) Demographics, (ii) Career Satisfaction and Mentorship, (iii) Workplace Culture and Discrimination, and (iv) Suggestions for Improvement (Appendix A). It included multiple-choice (with multi-selection options), open-ended questions for thematic analysis (e.g., improving oncology hiring practices), rating scales (e.g., rating workplace climate on cooperation, support, and diversity), and yes/no questions. “Culture of respect” was defined as colleagues’ attitudes, behaviors, and standards related to inclusion, access, and respect for diverse backgrounds and identities. Discrimination was defined as unjust treatment based on traits like race, age, gender, or sex. The term 2SLGBTQIA+ is an acronym used in Canada that stands for Two-Spirit, lesbian, gay, bisexual, transgender, queer or questioning, intersex, asexual, and the plus sign, which includes other identities. The addition of “2S” (Two-Spirit) is a Canadian-specific modification to be more inclusive of Indigenous people. The full acronym represents diverse gender and sexual identities that have historically faced oppression.

The survey was created in tandem with a trainee-focused survey and was delivered in English and French to encourage survey participation in either one of Canada’s two official languages (Appendix A, [21]).

### 2.3. Data Handling and Analysis

All data was stored securely on Qualtrics’ (https://www.qualtrics.com/, Seattle, WA, USA) online survey platform, and downloaded data was anonymous as no identifying information was collected or stored (such as IP address, institution, name, or email address). Any sensitive data that could be potentially identifiable due to a low response number was instead presented in aggregate to preserve anonymity.

Descriptive statistics were used to summarize the survey data. Survey questions were analyzed individually to incorporate responses by individuals who did not finish the entire survey or chose not to answer certain questions, but completed parts of it. Certain questions allowed participants to “multi-select” options if appropriate, and thus, response numbers are higher than the total participant numbers for those questions. Canadian population-level 2021 census data were obtained for relevant comparisons where the data exist and were supplied by Statistics Canada [22]. Canadian RO-specific data were obtained from the Canadian Institute for Health Information’s (CIHI) Scott’s Medical Database [23]. Chi-square tests were used to analyze between-group differences in race and gender. A *p*-value < 0.05 was deemed statistically significant.

### 2.4. Survey Response

Of the estimated 598 practicing Canadian ROs identified by the Canadian Institute of Health Information (CIHI) [24], 289 provided consent to participate in the survey, for a survey response rate that was 49.8% (with no response from the remaining 309 ROs). Of these, 273 answered at least one survey question (with 16 respondents quitting the survey without choosing any answers). There were 265 respondents who completed the entire sociodemographics section, and 254 (87.9%) respondents completed the entire survey. There was a final complete survey response rate of 42.5% (254/598) of all ROs (Figure 1). For some questions, respondents had the ability to select as many responses as they felt applicable, which is why response numbers vary per question.

### 2.5. Ethics Approval

The study was approved by our Institutional Research Ethics Board (Health Research Ethics Board of Alberta—HREBA.CC-22-0145).

## 3. Results

### 3.1. Demographics

Demographic data included questions regarding the following: gender identity, age, sexual orientation, race/ethnicity, religion, geographic location of practice, citizenship status, marital status, dependency status, disability status, primary language spoken, degrees earned, years of practice, academic rank, socioeconomic background, and scholarly work.

The gender and geographic distribution of respondents were similar to existing CIHI data for ROs and aligned with 2021 Canadian Census data (Table 1, *p* = 0.54) [24]. Most respondents practiced in Ontario (35.8%; 95/265) or Quebec (25.6%; 68/265). The majority were men (62.9%; 168/267), aged 35–44 (39.2%; 107/273), married or in a domestic relationship (85.4%; 228/267), and heterosexual (94.3%; 246/261). English was the primary language for 59.5% (157/264), though respondents spoke over 32 languages. Gender distribution was consistent with previous studies (62.9% male in this study and 63% male in past studies), but women were underrepresented compared to the general Canadian population (Table 1, *p* < 0.001) [25].

Regarding racial identity, 58.8% of respondents did not identify as a visible minority, while 41.2% (105/255) were racialized, primarily of Chinese (15.3%; 39/255) and South Asian (12.2%; 31/255) origin. Compared to the Canadian population, Caucasian ROs were underrepresented (58.8% vs. 70%), while racialized ROs were overrepresented (41.2% vs. 27%, *p* < 0.001). However, Black, Indigenous, and Southeast Asian ROs were underrepresented (1.9% vs. 4%, <1% vs. 5% and 1.6% compared to 4%, respectively) compared to their proportions in the general population.

A subset Chi-square analysis of race and gender revealed that of racialized ROs, only 20% (21/105) were women, whereas White women comprised 49.3% (74/150) of White ROs (Table 1, Chi-squared for Caucasian women vs. racialized women = 22.7, *p* < 0.001). There were four racialized male ROs for every racialized female RO (Figure 2). Regarding sexual orientation, 3.8% (10/261) of respondents identified as gay and 1.9% (5/261) identified as bisexual or as asexual.

Most ROs (96.2%; 255/265) were Canadian citizens (67.2% by birth and 29.1% by immigration). The majority of ROs identify as Christian (40.5%; 107/264) or had no religious affiliation (40.1%; 106/264). When asked if the respondent had a disability, 3.7% (10/270) replied “yes,” with deaf/hearing impairment, mobility/physical impairment, and mental health being the top disabilities reported.

### 3.2. Career Satisfaction and Mentorship

The second part of the survey focused on career satisfaction and mentorship (Table 2). Most respondents (75.4%; 187/248) were satisfied with their job. Although not statistically significant (Chi-squared = 1.745, *p* = 0.18), more male ROs (13.1%; 22/168) than female ROs (8.1%; 8/99) were unsatisfied with their job, compared to those who were satisfied. Significantly more racialized ROs (17.1%; 18/105) compared to White ROs (8%; 12/150) reported dissatisfaction with their job, compared to those who were satisfied, however (Chi-squared = 4.563, *p* = 0.026). Overall, 80.5% (211/262) had considered moving institutions, and 9.2% (24/262) often or very often regretted becoming a physician. Most respondents lacked formal mentorship: 58.8% (153/260) said no program existed in their department, 55% (143/260) did not mentor others, and 75.2% (194/258) did not have a mentor. The absence of mentorship was similar across gender (73.2% male, 71.7% female) and race (76.7% White, 75.2% racialized). Only 17.7% (46/260) agreed that having a mentor with similar demographic characteristics was important; 44.6% (116/260) were neutral.

### 3.3. Workplace Culture and Experiences of Discrimination or Harassment

The third part of the survey aimed to understand how ROs felt about the culture/environment of their oncology center and characterize their lived experiences of harassment/discrimination (Table 3). The “culture of respect” in respondents’ departments was reported as good to excellent by 76% (200/263), with only 11.4% (30/263) of respondents rating their departments as poor or very poor. However, 42.1% of respondents (106/252) reported experiencing discrimination in the past 5 years, with 21.4% (54/252) of respondents being discriminated against 2–4 times, 9.9% (25/252) 5–10 times, and 7.5% (19/252) of respondents reporting experiencing discrimination on a regular and ongoing basis. The majority of those discriminated against listed gender (25.9%; 52/201), race/ethnicity (20.4%; 41/201), and age (15.4%; 31/201) as the primary reasons for discrimination; however, pregnancy/caregiving burden (9.4%; 19/201), national origin (6.5%; 13/201), marital status (4%; 8/201), level of education (3.5%; 7/201), and political views (3%; 6/201) were also top reasons listed. Fellow faculty members and patients/family members were the main sources of discriminatory behavior (31.7% for each; 58/183), but administrative or non-faculty staff was also a top source listed (13.7%, 25/183). Of respondents who experienced discrimination by a faculty member, 19.1% (48/251) reported that the staff member was in a position of power and that they could directly affect the respondent’s academic or professional opportunities. Of respondents who experienced discrimination by a patient or the patient’s family member, race/ethnicity (35.8%; 43/120) was the number one reason cited, followed by gender (24.2%; 29/120) and age (22.5%; 27/120).

A large number (45%; 114/252) of respondents either felt uncomfortable reporting incidents or did not understand how to. Subset analyses found that female and male ROs report significantly different sources of discrimination (Chi-squared = 39.2, *p* < 0.0001). Female ROs overwhelmingly reported gender as a source of discrimination more than male ROs (44/52; 84.6% vs. 7/47; 14.9%), whereas male ROs meaningfully reported race/ethnicity more than female ROs as a reason for discrimination (30/47; 63.8% vs. 12/52; 23.1%). Female ROs also reported age (46.1%, 24/52), pregnancy/caregiver responsibilities (26.9%; 14/52), and marital status (13.5%; 7/52) as other top reasons of discrimination, whereas male ROs reported these factors less frequently. Racialized ROs were twice as likely to significantly report discrimination experiences compared to White ROs (60.4%; 67/111 vs. 29.5%, 44/149, chi-squared = 24.713, *p* < 0.001).

Types of anti-discrimination training provided to ROs were also evaluated. While the most common forms of training were via an online module or via a review of institutional policy, 43.9% (119/271) had no 2SLGBTQIA+ training, 35.6% (99/278) had no learner mistreatment training, 30.8% (82/266) had no anti-racism training, 24.6% (68/276) reporting no sexual harassment training, 24.5% (69/282) had no equity, diversity, and inclusion (EDI) training, and 17.5% (50/286) had no Aboriginal/Indigenous-specific population training.

### 3.4. Thematic Analysis

Respondents were asked, via open-text responses, how oncology departments should address mistreatment or harassment. Four key themes emerged: (i) training and education, (ii) clear reporting policies (including anonymous and structured complaint pathways), (iii) meaningful repercussions for mistreatment, and (iv) fostering safe environments that encourage reporting (Table 4). Some radiation oncologists (ROs) emphasized the need for mistreatment-focused training, accessible resources, and discussions at departmental meetings. Others suggested in-person sessions to clarify policies and the implementation of standardized, anonymous reporting mechanisms administered by an independent body. Respondents stressed the importance of thorough investigations and zero-tolerance policies. Many also called for cultural change and more diverse leadership to create safer, more inclusive workplaces. A minority (5.2%) believed no EDI-related issues exist in Canadian radiation oncology and that no changes are needed. When asked how to advance an EDI culture, respondents emphasized two themes: (i) support for underrepresented individuals to attain leadership roles and (ii) mandatory, comprehensive EDI education and training (Table 4). Many comments advocated for leadership development programs for women and minoritized groups, citing underrepresentation in senior roles. Others noted a general lack of formal EDI training opportunities. Respondents were asked how oncology departments could make faculty hiring fairer. Three main themes emerged: (i) adopting transparent, inclusive, and equitable hiring practices; (ii) ensuring diverse hiring committees; and (iii) hiring more candidates from outside their own residency programs (Table 4). Respondents emphasized the need for formal, transparent EDI-focused hiring processes to reduce barriers for underrepresented applicants. They also supported mandatory training for hiring committee members and greater external hiring to counter “group think” and foster constructive change.

## 4. Discussion

This study is the first to analyze and evaluate the demographics of Canadian ROs; gauge Canadian RO career satisfaction, workplace culture, and lived experiences of discrimination and harassment; and recognize areas of improvement within the field. Although most respondents reported a culture of respect and satisfaction with their job, discrimination was reported by almost half of the respondents. Discrimination can lead to physicians experiencing lower self-esteem, poorer self-care, higher rates of anxiety, depression, and family discord, decreased focus at work, increased susceptibility to substance abuse and suicidal ideation, and, ultimately, physician burnout and radiation oncologists leaving the workforce [26].

Despite overall job satisfaction, nearly half of respondents reported experiencing discrimination. Key associations revealed that women are underrepresented in radiation oncology (37.1%) compared to the Canadian labor force (49%), medical school entrants (59%), and active physicians (43%), though representation is higher than in the U.S. (36% vs. 26%) [22,25,27,28,29]. Racialized Canadian female ROs were seen as significantly underrepresented, comprising only 20% of racialized ROs, unlike White ROs, where gender distribution is nearly equal (50.7% men vs. 49.3% women, *p* < 0.001). Women were significantly more likely to report experiencing gender-based discrimination, while men reported race/ethnicity discrimination, and racialized ROs were twice as likely (29.5% vs. 60.4%) to report discrimination than White ROs. This is similar to the survey results from the Brazilian Society of Clinical Oncology, in which high rates of gender-based discrimination were reported by their female members (70.5% of women versus 1.8% of men) [30]. These associations can highlight the need for targeted mentorship, family-friendly policies, wellness programs, and inclusive recruitment strategies to improve diversity in radiation oncology.

We also identified that certain underrepresented races/ethnicities were present in the Canadian RO workforce (First Nations, Black, Southeast Asians) who may benefit from targeted pathway or mentorship programs. For example, the University of Toronto’s Black Student Application Program, launched in 2018, increased Black medical student representation from a single student in 2016 to 24 in 2020 (which was 9% of the class, compared to 7.5% of Toronto’s population) [31,32]. Similarly, in the U.S., Black, Hispanic, Indigenous, and Pacific Islander ROs remain significantly underrepresented, with Black ROs comprising only 3% of the workforce versus 13% of the general population [29]. Most respondents (82.3%; 214/260) felt neutral or disagreed that having a mentor with similar characteristics was important; however, this means that mentees may be open to a wider variety/pool of potential mentors.

The high rate of reported discrimination (42.1%) and the fact that 45% of respondents were unaware of or uncomfortable with reporting harassment are points of concern. Similar findings were seen in a U.S. medical physics study, where the American Association of Physicists in Medicine (AAPM) found that 38% of respondents experienced at least one type of discrimination [33]. The ESTRO diversity, equity, inclusion, and workforce engagement study also revealed that minority members reported low inclusion scores on access to opportunity, cultural competence, and respect, while the largest difference between minority and non-minority respondents was on “trust”. These results may underscore the need for supportive and respectful environments in regard to diversity, clear anti-discrimination policies, and effective training [6]. However, punitive measures alone are insufficient and should be paired with strategies to address underlying prejudice and bias [34]. Discrimination is a complex, multi-factor phenomenon, but research shows that strong institutional support can shift public perceptions of social norms and can increase acceptance of marginalized groups, as individuals often use perceptions of what is common or accepted in a collective as a guide to their own behavior. For example, support for same-sex marriage increased after the 2015 U.S. Supreme Court ruling [35]. And while existing research shows that intergroup prejudices are deeply ingrained, studies show that brief, empathy-driven conversations can reduce prejudice. Broockman and Kalla found that when people were encouraged to adopt a transgender person’s perspective, their support for anti-discrimination laws rose significantly [36]. Anonymous reporting systems with clear, measured outcomes (such as reporting the overall number of reports and how quickly they were addressed) may help ROs report harassment, especially since 31.7% experienced harassment from colleagues, with 19.1% reporting that the harassment was by someone in a position of power. Still, reporting systems alone may be ineffective if institutional culture does not support meaningful change [37]. Current anti-discrimination training is often inadequate, especially when delivered online.

Discrimination was most commonly reported from fellow faculty and patients or their families, with gender, race/ethnicity, and age cited as the top reasons. Similar patterns were seen in a study of medical physicists, where women faced gender-based barriers related to family planning, childcare, and career advancement [38]. In terms of patient or family members of patients who perpetrated discriminatory behavior, social identity-based discrimination against healthcare providers is a prevalent and well-documented phenomenon that needs to be addressed at a systems and institutional level to hold patients accountable [39].

Limitations of our study include the possibility of selection or sampling bias (as respondents needed to be reached via our established e-mail communication lists), recall bias as responses for discrimination or harassment was limited to a 5-year window, low numbers of certain respondent groups that limits the generalizability of certain analyses, and survey constraints including the survey design that may limit the breadth of responses along with a lack of ability to probe answers further. With a survey that deals with sensitive topics such as ours, non-responders who are unwilling to answer may also reflect a population differing from those who do. Therefore, generalizing these results to the entire Canadian RO workforce has to be carried out with caution. Still, 42.5% of Canadian ROs completed the survey in full, which is a strength, and the geographic distribution of responses aligns with national CIHI data.

## 5. Conclusions

This survey summarizes the sociodemographic information, workplace environment, rates of discrimination and harassment, and details areas of improvement within the field of radiation oncology in Canada. Results show that women and certain racialized minorities, such as First Nations, Black, and Southeast Asian, are underrepresented. Data also identifies areas of concern related to high levels of RO mistreatment and discrimination (particularly amongst women and racialized ROs) and provides suggestions from ROs themselves for improvement. The findings in this study suggest that the creation of targeted mentorship/pathway programs, physician wellness plans, diversity initiatives, and the construction of more clearly defined, actionable, transparent, and robust institutional policies to tackle mistreatment and discrimination can help advance EDI within the field of radiation oncology in Canada.

## Figures and Tables

**Figure 1 curroncol-32-00643-f001:**
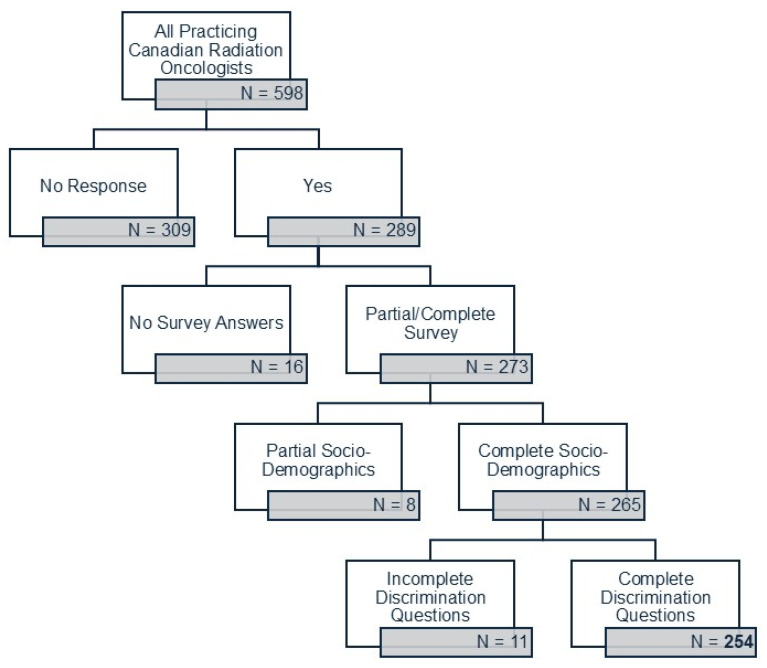
Flowchart of survey responses.

**Figure 2 curroncol-32-00643-f002:**
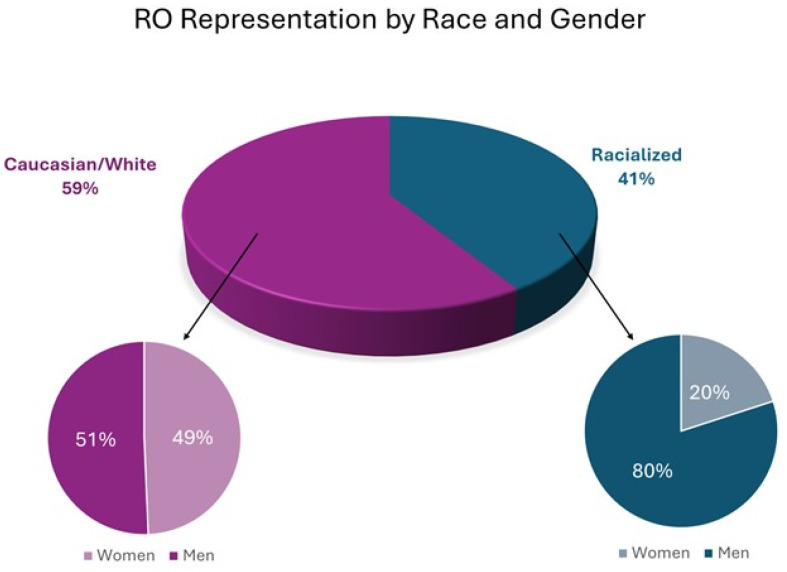
Radiation oncology staff representation by race and gender.

**Table 1 curroncol-32-00643-t001:** Demographic data of respondents to the survey.

Demographic Category (Number of Respondents)	Demographic Characteristic	Survey—RO Data	2022 CIHI Data—RO Population ^1^	2021 Census Data ^2^	Significance/Notes
**Gender Identity** (267)	Man Woman	168 (63%) 99 (37%)	380 (64%) 218 (36%)	12.9 m (48%) 13.8 m (52%)	
**Age** (273)	25–34 35–44 45–54 55–64 65+	15 (5%) 107 (39%) 82 (30%) 47 (17%) 22 (8%)	-	4.4 m (22%) 4.4 m (22%) 4.0 m (20%) 3.3 m (16%) 95 m (4%)	Census data only included for working persons, not the general population
**Sexual Orientation** (Multiselect, 261)	Heterosexual Gay Bisexual Asexual Queer Pansexual Questioning	246 (94%) 10 (4%) 4 (2%) 1 (<1%) 0 0 0	-	-	
**Race/Ethnicity** (Multiselect, 255)	Caucasian/White Racialized Group *Chinese* *South Asian* ^4^ *Black* *Southeast Asian* ^5^ *First Nations* *Pooled Others*	150 (59%) 105 (41%) 39 (15%) 31 (12%) 5 (2%) 4 (1%) 1 (<1%) 25 (10%)	-	25 m (70%) 9.6 m (27%) 1.7 m (5%) 2.5 m (7%) 1.5 m (4%) 1.3 m (4%) 1.8 m (5%) 2.5 m (7%)	
**Race and Gender—Subset Analysis**	Caucasian Overall Caucasian Male RO Caucasian Female RO	150 (59%) 76 (51%) 74 (49%)	-	25 m (70%)	Chi-squared for Caucasian women vs. racialized women = 22.7, *p* < 0.001
	Racialized Overall Racialized Male RO Racialized Female RO	105 (41%) 84 (80%) 21 (20%)	-	9.6 m (27%)	
**Religion** (Multiselect, 264)	Christianity Atheist/No religion Hinduism Islam Spiritual Judaism Buddhism Sikhism Other	107 (41%) 106 (40%) 12 (5%) 11 (4%) 11 (4%) 11 (4%) 3 (1%) 1 (<1%) 2 (<1%)	-	19.3 m (53%) 12.6 m (34%) 0.83 m (2%) 1.7 m (5%) 80 k (<1%) 0.36 m (1%) 0.34 m (1%) 0.77 m (2%) 229 k (<1%)	
**Are you identifiable as a member of a religion?** (262)	Definitely yes Probably yes Probably no Definitely no Not sure	17 (6%) 23 (9%) 75 (29%) 133 (51%) 14 (5%)	-	-	
**What religion would people assume you belong to?** (Multiselect, 252)	Christianity Hinduism Buddhism Islam Judaism Confucianism Sikhism Atheist/Agnostic	159 (63%) 20 (8%) 19 (8%) 19 (8%) 18 (7%) 8 (3%) 5 (2%) 3 (1%)	-	-	
**Geography** (265)	Ontario Manitoba and the West Quebec Maritimes	95 (36%) 85 (32%) 68 (26%) 17 (6%)	233 (39%) 179 (30%) 143 (24%) 43 (7%)	10.3 m (38%) 8.4 m (31%) 6.2 m (23%) 2.0 m (8%)	
**Citizenship Status** (265)	Canadian *By birth* *By immigration* PR Work Visa	255 (96%) 178 (67%) 77 (29%) 6 (2%) 4 (1%)	-	33.1 m (91%) 27 m (74%) 6.1 m (17%) - -	
**Marital Status** (267)	Married/Domestic Single Widowed Divorced/Separated	228 (85%) 26 (10%) 4 (1%) 9 (3%)	-	4.9 m (57%) 3.7 m (43%) - -	
**How many dependents do you have?** (273)	01 2 3+	78 (29%) 42 (15%) 79 (29%) 74 (27%)	-	-	
**Primary language** (264)	English French Another language	157 (59%) 54 (20%) 53 (20%)	-	27.8 m (76%) 8 m (22%) 0.67 m (2%)	
**Degrees earned** (Multiselect, 273)	MD Masters PhD MBA JD	273 (100%) 89 (33%) 32 (12%) 10 (4%) 2 (1%)	-	-	
**Where MD was obtained** (273)	Canada International	220 (81%) 53 (19%)	454 (76%) 144 (24%)	-	
**Residency Training Location** (273)	Canada Elsewhere	239 (88%) 34 (12%)	-	-	
**Years of Practice** (273)	<5 6–10 11–15 16–20 21–25 26+	41 (15%) 58 (21%) 66 (24%) 24 (9%) 37 (14%) 47 (17%)	-	-	
**Academic Rank** (273)	Assistant Professor Associate Full Professor Lecturer No Appointment	110 (40%) 81 (30%) 41 (15%) 25 (9%) 16 (6%)	-	-	
**How large is your practice group?** (273)	1–5 6–10 11–20 21–30 30+	16 (6%) 41 (15%) 113 (41%) 68 (25%) 35 (13%)	-	-	
**Do parents have a degree(s)?** (273)	One Both Neither No answer	57 (21%) 117 (43%) 94 (34%) 5 (2%)	-	-	
**Income when growing up** (235)	CAD 150,000+ CAD 100,000–CAD 150,000 CAD 50,000–CAD 100,000 CAD 25,000–CAD 50,000 <CAD 25,000 I don’t know No answer	54 (23%) 38 (16%) 76 (32%) 28 (12%) 21 (9%) 3 (1%) 15 (6%)	-	-	
**How many peer-reviewed publications have you been an author on?** (260)	<5 5–10 10–25 25–50 50–100 >100	59 (23%) 49 (19%) 52 (20%) 40 (15%) 26 (10%) 34 (13%)	-	-	
**Do you view yourself as having a disability?** (270)	Yes No Prefer not to answer	10 (4%) 256 (95%) 4 (1%)	-	27% ^3^ 73% -	
**What is your disability?** (multi-select, 16)	Deaf/hearing Mobility/physical Mental health Autism Cognitive Chronic illness Speech Other	3 (19%) 3 (19%) 3 (19%) 2 (13%) 1 (6%) 1 (6%) 1 (6%) 2 (13%)	-	-	

^1^: Canadian Institute for Health Information. Supply, Distribution and Migration of Physicians in Canada, 2022—data tables. Ottawa, ON: CIHI; 2023. https://www.cihi.ca/en/scotts-medical-database-metadata (accessed on 7 December 2024); ^2^: Statistics Canada—Data tables for the 2021 Census of Population. Ottawa, ON, 2021. https://www12.statcan.gc.ca/census-recensement/2021/dp-pd/dt-td/index-eng.cfm (accessed on 12 May 2024); ^3^: Statistics Canada—Data tables for the 2022 Canadian Survey on Disability. Ottawa, ON, 2022. https://www23.statcan.gc.ca/imdb/p2SV.pl?Function=getSurvey&SDDS=3251 (accessed on 24 March 2024); ^4^: Comprised of persons from India, Bangladesh, Pakistan, Sri Lanka, Nepal, Bhutan, and Maldives, and can include people of South Asian ancestry who historically immigrated to places like Trinidad and Tobago, Guyana, and East/South Africa; ^5^: Comprised of persons from Cambodia, Indonesia, Lao, Malaysia, Singapore, Thailand, and Vietnam.

**Table 2 curroncol-32-00643-t002:** Career satisfaction and mentorship status of respondents to the survey.

Question (Number of Respondents)		Study Sample	Significance/Notes
**“All in all, I feel satisfied with my job”** (248)	Strongly disagree Disagree Neither agree nor disagree Agree Strongly agree	14 (6%) 16 (6%) 31 (13%) 114 (46%) 73 (29%)	
**“All in all, I feel satisfied with my job”—subset analysis by gender of those picking “Disagree” or “Strongly Disagree”** (30, percentage is of overall gender)	Male Female	22 (13%) 8 (8%)	Chi-squared comparing gender vs. satisfaction = 1.745, *p* = 0.18
**“All in all, I feel satisfied with my job”—subset analysis by race of those picking “Disagree” or “Strongly Disagree”** (30, percentage is of overall race)	Caucasian/White Racialized	12 (8%) 18 (17%)	Chi-squared comparing race vs. satisfaction = 4.563, *p* < 0.026
**How often have you thought about moving to a different institution?** (262)	Never Once or twice Sometimes Often Very often	51 (19%) 83 (32%) 87 (33%) 23 (9%) 18 (7%)	
**How often have you felt regret about deciding to become a physician?** (262)	Never Once or twice Sometimes Often Very often	131 (50%) 50 (19%) 57 (22%) 16 (6%) 8 (3%)	
**A formal mentorship program exists within my department** (260)	Yes No	107 (41%) 153 (59%)	
**I currently act as a mentor to a trainee(s) and/or colleague(s)** (260)	Yes, trainee(s) Yes, colleague(s) Yes, trainee(s) and colleague(s) No	62 (24%) 22 (8%) 35 (13%) 143 (55%)	
**I currently have at least one mentor** (258)	Yes No	64 (25%) 194 (75%)	
**I currently have at least one mentor—subset analysis by gender for “No” responses** (194, percentage is of overall gender)	Male Female	123 (73%) 71 (72%)	
**I currently have at least one mentor—subset analysis by race for “No” responses** (194, percentage is of overall race)	Caucasian/White Racialized	115 (77%) 79 (75%)	
**It is important I have a mentor with similar demographic characteristics to me** (260)	Strongly disagree Disagree Neither agree nor disagree Agree Strongly agree	33 (13%) 65 (25%) 116 (45%) 40 (15%) 6 (2%)	

**Table 3 curroncol-32-00643-t003:** Workplace culture, discrimination, and harassment experience of respondents to the survey.

Question (Number of Respondents)		Study Sample	Significance/Notes
**Thinking about the past year, how would you rate the culture of respect in your department?** (263)	Excellent Very good Good Adequate Poor Very Poor	59 (22%) 87 (33%) 54 (21%) 33 (13%) 20 (8%) 10 (4%)	
**How often did you feel that you experienced discrimination in the past 5 years while working as a radiation oncologist?** (252)	Never Once 2–4 times 5–10 times Regularly/ongoing basis	146 (58%) 8 (3%) 54 (21%) 25 (10%) 19 (8%)	
**In the past 5 years as an RO on what basis have you felt discriminated upon?** (Multiselect, 201)	Gender Race/ethnicity Age Pregnancy/caregiving National origin Marital status Level of education Political view Disability Religion Language Sexual orientation Socioeconomic status Lack of research interest Seniority IMG status Caring for elderly parents Traveling during COVID-19 Lack of children Medical specialty	52 (26%) 41 (20%) 31 (15%) 19 (9%) 13 (6%) 8 (4%) 7 (3%) 6 (3%) 4 (2%) 4 (2%) 5 (2%) 3 (1%) 1 (<1%) 1 (<1%) 1 (<1%) 1 (<1%) 1 (<1%) 1 (<1%) 1 (<1%) 1 (<1%)	
**What was the role of the person(s) who discriminated against you?** (Multiselect, 183)	Faculty member Patient/Patient’s family Staff (administrative, non-faculty) Other allied health professional Nurse Resident/Clinical Fellow Funding agencies	58 (32%) 58 (32%) 25 (14%) 19 (10%) 16 (9%) 6 (3%) 1 (<1%)	
**Was the person who discriminated against you someone in a position to directly affect your academic, and/or professional opportunities?** (251)	Yes No Not sure Does not apply No answer	48 (19%) 52 (21%) 7 (3%) 138 (55%) 6 (2%)	
**If you experienced harassment/discrimination perpetrated by a patient/family at any time during your career, on what basis did they discriminate against you?** (Multiselect, 120)	Race/ethnicity Gender Age National origin Other Sexual orientation Disability Religion	43 (36%) 29 (24%) 27 (23%) 10 (8%) 8 (7%) 1 (<1%) 1 (<1%) 1 (<1%)	Chi-squared comparing sources of discrimination by gender = 39.2, *p* < 0.001
**Women and Reported Discrimination Reasons—Subset Analysis** (Multiselect, 52)	Gender Age Pregnancy/Caregiver Responsibilities Race/ethnicity Marital Status Other	44 (85%) 24 (46%) 14 (27%) 12 (23%) 7 (13%) 5 (10%)	
**Men and Reported Discrimination Reasons—Subset Analysis** (Multiselect, 47)	Race/ethnicity Age Gender Childcare/Caregiver Responsibilities Marital Status Other	30 (64%) 9 (19%) 7 (15%) 4 (8.5%) 1 (2%) 14 (30%)	
**Ethnicity and Reporting at Least One Type of Discrimination—Subset Analysis**	White Racialized	44/149 (30%) 67/111 (60%)	Chi-squared = 24.713, *p* < 0.001
**“I understand how to and feel comfortable reporting harassment incidents at my workplace”** (252)	Strongly disagree Somewhat disagree Neither agree/disagree Somewhat agree Strongly agree	39 (15%) 75 (30%) 44 (17%) 66 (26%) 28 (11%)	
**I have training regarding sexual harassment** (multiselect, 276)	No training In person Online module Review of institutional policy	68 (25%) 10 (4%) 149 (54%) 49 (18%)	
**I have training regarding racism** (multiselect, 266)	No training In person Online module Review of institutional policy	82 (31%) 9 (3%) 127 (48%) 48 (18%)	
**I have training regarding 2SLGBTQIA+ peoples** (multiselect, 271)	No training In person Online Review of institutional policy	119 (44%) 10 (4%) 95 (35%) 47 (17%)	
**I have training regarding Aboriginal/Indigenous health** (multiselect, 286)	No training In person Online module Review of institutional policy	50 (17%) 26 (9%) 164 (57%) 46 (16%)	
**I have training regarding learner mistreatment** (multiselect, 278)	No training In person Online module Review of institutional policy	99 (36%) 23 (8%) 105 (38%) 51 (18%)	
**I have training regarding equity, diversity and inclusion** (multiselect, 282)	No training In person Online module Review of institutional policy	69 (24%) 18 (6%) 145 (51%) 50 (18%)	

**Table 4 curroncol-32-00643-t004:** Thematic content and select quotes from open-text questions regarding how oncology centers can improve.

Question	Select Summarized and Representative Quotes
**What should oncology departments do to address mistreatment or harassment?**	(i) Training/education: *“Better training with easily accessible resources. Publicize them to make them known to everyone.* *“Open discussions at the department meetings. Educational workshops. Form special committees.”* (ii) Having clear institutional policies on how to report mistreatment (anonymously or via a structured complaint pathway): *“Have open in-person conversations led by leadership to clarify expectations, policies and procedures, and have a forum to confidentially report mistreatment/harassment to for victims and an avenue for understanding options for formal reporting.”* *“A standardized / easy to access / anonymous reporting method is needed. There should be an independent process/body to administrate this to reduce bias.”* (iii) Having meaningful and measured corrective measures or repercussions to mistreatment: *“Incidents should be fully investigated and sorted out and a clear message of no tolerance for any form of mistreatment or harassment should be in place.”* *“We need more than investigation of just the incident. There needs to be an assessment of the work environment to determine if this is an isolated behaviour or whether more widespread issues exist. Mistreatment and harassment should not be tolerated in any form.”* (iv) Creating safe environments where reporting mistreatment or harassment is encouraged: *“Explore the cultural factors that are leading to the issues and be willing to significantly disrupt culture that is perpetuating these factors. There needs to be a way to report incidents in a confidential and safe manner.”* *“Need more diversity in leadership. Most leaders are one gender and appear the same, we need to hire people from multiple backgrounds that represent all ROs to be leaders/administration.”*
**What should oncology departments do to advance equity diversity and inclusion (EDI) in the workplace?**	(i) Advancement supports for underrepresented persons in RO to achieve leadership positions: *“Support leadership training for underrepresented faculty and trainee members.”* *“Stop looking for a certain archetype as this creates a selection bias. We need to clear policies for promotion to hire people in leadership positions with different skills and backgrounds.”* (ii) Mandatory and comprehensive EDI education and training: *“There needs to be in-person mandatory presentations to learn what EDI is and how it may manifest, clear training on policies and institutional goals related to EDI, with discussion around how to mitigate barriers to EDI goals.”* *“We need to educate ourselves on best practices and make conscious efforts to advance and seek expertise to improve EDI training.”*
**What should oncology departments do to make faculty hiring practices more equitable?**	(i) Transparent, inclusive, and equitable hiring practices: *“We need better diversity in hiring practices, and teaching on why this is valuable. Departments need to make this a priority and openly discuss how they are taking the lead on improving recruitment processes and mitigating unconscious biases.”* *“There needs to be formal processes and hiring practices, informed by EDI experts, that supports the development of a diverse work force and decrease barriers for those who are underrepresented with inclusive language.”* (ii) Ensuring that hiring committees are diverse themselves: *“Committees should be composed of a diverse group of individuals to make hiring decisions. Members should have mandatory training on hiring policies and best practices.”* *“Follow established university hiring practices that already focus on EDI and fairness and seek advice from hospital/university affiliates to ensure that the committee and department hiring is open to equity and diversity and hiring members are from diverse backgrounds.”* (iii) Hiring candidates externally/from a different training background: *“There should be an external review of any hire that confirms that a job posting was publicly available with specific considerations as to whether potential hires external to the department were given due consideration. The hiring of our own trainees can perpetuate the retention of a stagnant culture, which is reinforced when we are disrupted by an external hire (i.e., the culture conflict often makes everyone wish the hire was internal, when in fact the disruption of group think was a positive).*

## Data Availability

Due to the sensitive nature of this study and to protect the privacy of respondents’ personal information, supporting data are not available.

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
