# Peer review of "Diversity and Experiences of Radiation Oncologists in Canada: A Survey of Gender Identity, Sexual Orientation, Disability, Race, Ethnicity, Religion, and Workplace Discrimination—A National Cross-Sectional Electronic Survey"

_curroncol, 2025, doi:10.3390/curroncol32110643_

Round 1

Reviewer 1 Report

Comments and Suggestions for Authors

Hi- thank you for putting together an interesting and highly relevant research paper. My only critique is perhaps the lack of engagement with emotional or psychological consequences of facing discrimination- the impact on the radiologists quality of work or sense of self, it is very matter of fact and provides a lot of stats which is the area of relevance, but a paragraph a few sentences in the introduction or discussion would not go amiss. 

Author Response

Comment 1: Hi- thank you for putting together an interesting and highly relevant research paper. My only critique is perhaps the lack of engagement with emotional or psychological consequences of facing discrimination- the impact on the radiologists quality of work or sense of self, it is very matter of fact and provides a lot of stats which is the area of relevance, but a paragraph a few sentences in the introduction or discussion would not go amiss. 

Response 1: Thank you very much for your review and feedback. We edited the first paragraph of the discussion section with the following: "Discrimination can lead to physicians experiencing lower self-esteem, poorer self-care, higher rates of anxiety, depression and family discord, decreased focus at work, increased susceptibility to substance abuse and suicidal ideation and ultimately: physician burnout and radiation oncologists leaving the workforce."

We agree with your sentiment. We had an open-ended text box in our survey (with results not published due to the possibility of identification in our small field) where radiation oncologists could anonymously leave us comments and it was very upsetting to see how discrimination has personally impacted those in our field.

Reviewer 2 Report

Comments and Suggestions for Authors

This study explored the demographics and experiences of discrimination among Canadian radiation oncologists (ROs), from ethnicity, gender identity, disability, sexual orientation, and socioeconomic background aspectives. The comments are as follows.

1. The quality control of the survey should be explained in details. 

2. In Table 1,  "Italic font signifies significant statistical findings." (line 182-183), please  move it to the footnote of the table. 

3. Table 4, please relocate it to the results part. 

4. The references requires an in-depth revision of format to meet to the requirement of the journal.

Comments on the Quality of English Language

This study needs a moderate language correction. 

Author Response

Comment 1: The quality control of the survey should be explained in details. 

Response 1: Thank you for your comments and feedback. We changed the first paragraph of our Materials and Methods to expand on our quality control methods for the creation of the survey. It now reads as follows:

The study was approved by our Institutional Research Ethics Board. A 45-question survey was developed to gather anonymous cross-sectional data, as there was no validated survey specifically designed for radiation oncologists in regards to EDI. The survey creation process was informed by a literature review and input from equity, diversity and sociodemographic survey experts across Canada and the U.S. to ensure questions were inclusive, culturally sensitive, able-focused rather than disability-focused and when appropriate were drawn from validated survey sets in other fields of medicine. The concept of underrepresented and minoritized individuals was based on the AAMC’s definition of representation in medicine relative to the general population. A pilot survey was conducted at our local institution to identify confusing wording, leading questions and define clear inclusion and exclusion criteria to avoid non-representative responses.

Comment 2: In Table 1,  "Italic font signifies significant statistical findings." (line 182-183), please  move it to the footnote of the table. 

Response 2: This has been edited.

Comment 3: Table 4, please relocate it to the results part. 

Response 3: This has been edited, along with more explanation of the content of the table.

Comment 4: The references requires an in-depth revision of format to meet to the requirement of the journal.

Response 4: This has been performed with the comments also left by the journal editorial staff.

Reviewer 3 Report

Comments and Suggestions for Authors

The authors submit the manuscript “Diversity and Experiences of Radiation Oncologists in Canada: A Survey of Gender Identity, Sexual Orientation, Disability, Race, Ethnicity, Religion and Workplace Discrimination,” an interesting analysis of the reality of Canadian radiation oncologists. While the study reports valuable findings and addresses a knowledge gap, it is affected by numerous issues that must be resolved before it can be considered for publication.

Title 

  • The authors should add “national cross-sectional electronic survey” in the subtitle/Abstract to clearly state the study design.

Abstract

  • Standardize N and percentages; replace imprecise p-values such as “p<0.03” with exact values (e.g., p=0.028) and report effect sizes.

  • The Simple Summary and Abstract report 298/598 (48%), but the Results state 289 consented; 273 answered ≥1 item; 254 completed. Unify all figures across the text, abstract, figures, and legends.

  • The Simple Summary/Abstract state South Asian under-representation (1% vs 4%), but the table (and Results) show 12% South Asian. Correct the Summary and Abstract to reflect the actual data.

Introduction

  • Avoid repetition and redundant information. Ensure the final paragraph of the Introduction ends with a clear, explicit statement of the study objective.

Materials and Methods

  • This section lacks logical structure and several essential details. It should explicitly state: study design, setting, population, participant selection criteria, instrument design, data handling, statistical measures, and ethical considerations. Please review and restructure the entire section, following STROBE guidelines for cross-sectional studies, as applicable.

  • Ethics approval should be presented as a sub-section at the end of Materials and Methods.

Other observations (measurement quality):

  • “Culture of respect”: you define the construct but do not report internal validation. If multiple Likert items were used, provide Cronbach’s α and justify any dichotomization (“good–excellent” vs “adequate–very poor”). Consider reporting Likert means (±SD) in addition to categorized percentages.

  • Add a multivariable logistic regression with the outcome “reported ≥1 discrimination episode (yes/no)” and include predictors such as sex, racialized status, age, years in practice, province, academic rank, group size, and mentorship. This will yield adjusted ORs and identify independent predictors beyond bivariate comparisons. (The outcome is already defined: 42% reported discrimination in the past 5 years.)

Results

  • The section provides important information; however, please consider:

    • The response flow and rate (with a flow diagram) are useful, but the process description is better placed in Materials and Methods. Begin the Results with sample demographic characteristics.

    • Some items show N greater than the number of completed surveys (e.g., training items with N=286/282/278) and variable denominators elsewhere. In Methods, state that analyses were item-level, and in tables/figures show valid N per item. Ensure no N exceeds the number of valid respondents.

    • You use “LGBTQ2+” in one place and “2SLGBTQIA+” in another. Harmonize to 2SLGBTQIA+ and explain this terminology in Methods (aligned with Canadian guidance).

Discussion

  • Frame findings as associations, not causal inferences. Discuss selection bias (email lists, self-selection on sensitive topics) and recall bias (5-year window). Add comparison with ESTRO/AAPM studies for context (already cited). Strengthen institutional recommendations with measurable actions (e.g., “% of reports addressed within ≤30 days”).

Conclusions

  • Avoid starting with “In conclusion, this survey summarizes the sociodemographic information of ROs in Canada”—this is redundant. Conclusions should answer the study objective directly and highlight the actionable implications.

Author Response

Please see the attached word document for our detailed response to Reviewer 3, thank you.

Round 2

Reviewer 2 Report

Comments and Suggestions for Authors

The authors have revised the manuscript according to the reviewer's comments. Thus, the manuscript can be accepted after proofreading.

Reviewer 3 Report

Comments and Suggestions for Authors

No further comments.

The article ir aceptable in its current form.